# Potential Anti-Inflammatory Effect of *Rosmarinus officinalis* in Preclinical In Vivo Models of Inflammation

**DOI:** 10.3390/molecules27030609

**Published:** 2022-01-18

**Authors:** Catarina Gonçalves, Daniela Fernandes, Inês Silva, Vanessa Mateus

**Affiliations:** H&TRC—Health and Technology Research Center, ESTeSL—Lisbon School of Health Technology, Instituto Politécnico de Lisboa, 1990-096 Lisbon, Portugal; catarinagraisgoncalves@hotmail.com (C.G.); daniela.dsff@gmail.com (D.F.); ines.silva@estesl.ipl.pt (I.S.)

**Keywords:** inflammation, *Rosmarinus officinalis* L., rosmarinus, rosmarinic acid, carnosic acid, carnosol, rats, mice, mouse

## Abstract

This systematic review aimed to evaluate the potential anti-inflammatory effect of *Rosmarinus officinalis* in preclinical in vivo models of inflammation. A search was conducted in the databases PubMed, Scopus, and Web of Science, with related keywords. The inclusion criteria were inflammation, plant, and studies on rats or mice; while, the exclusion criteria were reviews, studies with in vitro models, and associated plants. The predominant animal models were paw edema, acute liver injury, and asthma. Rosemary was more commonly used in its entirety than in compounds, and the prevalent methods of extraction were maceration and hydrodistillation. The most common routes of administration reported were gavage, intraperitoneal, and oral, on a route-dependent dosage. Treatment took place daily, or was single-dose, on average for 21 days, and it more often started before the induction. The most evaluated biomarkers were tumor necrosis factor (TNF)-α, interleukin (IL)-1β, IL-6, IL-10, myeloperoxidase (MPO), catalase (CAT), glutathione (GSH), glutathione peroxidase (GPx), malondialdehyde (MDA), and superoxide dismutase (SOD). The best results emerged at a dose of 60 mg/kg, via IP of carnosic acid, a dose of 400 mg/kg via gavage of *Rosmarinus officinalis*, and a dose of 10 mg/kg via IP of rosmarinic acid. *Rosmarinus officinalis* L. showed anti-inflammatory activity before and after induction of treatments.

## 1. Introduction

Globally, therapeutic plants have been used by various communities, having a relevant role in the treatment of human and animal diseases. Today, they have been investigated increasingly often because of their benefits and fewer side effects when compared to pharmacological drugs. They can also be used as a complementary treatment to boost therapeutic progress [1].

*Rosmarinus officinalis* L., which belongs to the *Lamiaceae* family, is an aromatic evergreen plant with upright stems, whitish-blue flowers, and dark green leaves. It is commonly known as rosemary and is native in countries of the Mediterranean region. Fresh and dried leaves represent the most relevant part of the plant and can be used as a spice or to make herbal tea [2,3,4].

Rosemary’s chemical composition varies in different extracts, but its analysis shows that phenolic diterpenes, triterpenes, and phenolic acids are the most relevant active constituents. Regarding phenolic compounds, carnosic acid, carnosol, and rosmarinic acid, have been declared to have the main therapeutic effects, such as antioxidant, anti-inflammatory, antiviral, and antibacterial activities [2,3,4]. Plant extracts can be obtained from roots, stems, leaves, flowers, fruits, seeds, and bark, using selective solvents and standard procedures. Qualitative and quantitative studies on bioactive compounds isolated from plants depend on the proper selection of extraction method, which is a vital choice for obtaining satisfactory results [1,4].

The aerial parts of *Rosmarinus officinalis* have been widely used in different cultures as a food preservative and also as a flavoring agent in foods, beverages, and in cosmetics [2]. They are reported to have a variety of specific therapeutic properties, such as being hypoglycemic, antiatherogenic, antihypertensive, hypocholesterolemic, antioxidant, anti-inflammatory, hepatoprotective, antidepressant, antiproliferative, and antibacterial. It may also improve asthma, cataract, renal colic, peptic ulcer, and physical and mental fatigue [2,3,4].

Inflammatory diseases are widely known to be the main cause of morbidity across the global population. If inflammation is not controlled, it may result in numerous diseases, including rheumatoid arthritis, multiple sclerosis, inflammatory bowel disease, psoriasis, immune-inflammatory illnesses, and neoplastic transformations. Moreover, chronic inflammation is also associated with stages of tumorigenesis, presenting a risk factor for the occurrence of certain types of cancers. Chronic diseases tend to manifest as a sustained low-grade inflammation. In some of those diseases, treatment still represents a challenge, given the lack of safe and effective medications. As a response to the difficulties in finding safe and effective treatment options to control inflammation, many animal models have been developed to study and evaluate drug anti-inflammatory activities. To carry out these studies, the choice of the appropriate animal model for the preclinical experiment represents a challenge; in order to, afterward, establish the efficacy and translation of the drugs therapeutic properties in humans. Even though there are numerous in vivo models of inflammation, developed to access the potential of anti-inflammatory drugs, the proper selection of an animal model is always crucial. Unsuitable selection of animal models may lead to a false positive or false negative result and, therefore, prevent the identification of a possibly promising drug [5].

This study aimed to evaluate the potential anti-inflammatory effects demonstrated by *Rosmarinus officinalis* in preclinical in vivo models of inflammation, through a systematic review of the literature.

## 2. Results and Discussion

Applying the research expressions to Pubmed, Scopus, and Web of Science resulted in 338 studies, since 1994. Excluding duplicates and those for which it was not possible to access the full text, resulted in 216 studies for analysis. Some full-text articles were excluded because they did not meet the inclusion criteria, namely (1) without the plant *Rosmarinus officinalis* (*n* = 68); (2) without inflammation (*n* = 34); and (3) studies carried out in non-rodent animals (*n* = 4). Other full-text articles were excluded because they fit the exclusion criteria: (1) reviews, opinion articles, and clinical cases (*n* = 11); (2) exclusive studies with in vitro models (*n* = 34); and (3) studies with the associated plant (*n* = 2), as shown in Figure 1.

Data from the selected studies are compiled in Table 1. To gather all crucial data for further analysis, the information was simplified in columns, with parameters of interest for comparison. Therefore, the parameters screened in all reviewed articles were the animal model, the plant or compound studied and how it was extracted, dose and route of administration, frequency and duration of the treatment, and the biomarkers evaluated in the study.

### 2.1. Animal Model

In the analyzed studies, many models were used in rats and mice to test the anti-inflammatory activity of *Rosmarinus officinalis*. However, most models have only been used once, which makes comparison difficult. Even so, it is possible to highlight that paw edema model was the most used (*n* = 10), followed by models of acute liver injury and asthma (*n* = 4), and then, models of colitis, neuropathic pain, arthritis, ear edema, and hepatocarcinoma (*n* = 3). Some of these models are concordant with findings in the literature, namely, the models of paw edema [69,70,71,72,73,74], ear edema [71,75,76,77,78,79], and arthritis [69,74,80].

According to the model, the induction method differs. In this review, the method most used for paw edema was injection of carrageenan (*n* = 6); in the case of asthma, it was with ovalbumin (*n* = 3); and in colitis, it was mostly dextran sulfate sodium (*n* = 2). In acute liver injury, no method prevailed.

In the examined studies, the induction of inflammation predominant in paw edema models was an injection of carrageenan. These results are concordant with the literature because the paw edema model was not only the most used, but the preferred pathway of induction was also carrageenan [69,70,72,73,75,80]. The literature also showed the use of other pathways such as histamine [70,74,80], dextran [69,72,74], and serotonin [70,73,80]. In this review, besides carrageenan, all the other pathways of induction were only used once; therefore, making it unwise to compare.

The paw edema model is prevalent for assessing inflammation, probably because of its high reproducibility, and as it can be used as a preliminary test to screen potential anti-inflammatory drugs [5]. Models induced by carrageenan were widely investigated and used because of this substance’s ability to cause non-immune acute inflammation [5,81]. These models are essential for the development of drugs, and as a response to the inflammation induced by carrageenan the paw increases in size [81].

Although the ear edema model was one of the most investigated, in the analyzed studies and in the literature, all ear edema studies in this review used different induction pathways, which makes comparison difficult. In the literature, the most used methods of induction were oxazolone [71,76,77], 12-O-tetradecanoilforbol acetate (TPA) [78,79,82], ethyl phenylpropionate [75,83], and arachidonic acid [71,78]. However, in one study, inflammation was induced by croton oil, which is the irritant principle of TPA [5]. Ear edema models are valuable for topically assessing the anti-inflammatory and antioxidant potential of plant extracts. In addition, they also assess for steroidal and non-steroidal anti-inflammatory drug activity. TPA-induction inflammation increases cell proliferation and arachidonic acid metabolism in epidermal cells and generates a thickening of the skin 4 h after induction.

### 2.2. Plant/Compound and Extraction

Regarding the plant, *Rosmarinus officinalis*, it can be used in its entirety or in the form some of its isolated compounds. In the studies considered in this review, the plant was mostly used in its entirety (*n* = 31). However, rosmarinic acid, one of its compounds, was also extensively used (*n* = 26). Other compounds, such as carnosic acid (*n* = 12) and carnosol (*n* = 7) were less tested.

Studies with some of the main compounds present in *Rosmarinus officinalis* have been increasing in recent decades. Since 1990, 84 studies about carnosic acid, 46 studies about carnosol, and 32 studies about rosmarinic acid have been published [1]. The literature corroborates the compounds approached in the studies in this review. Although, rosmarinic acid is the compound with the least published studies and in this review, it was the most studied compound after the whole plant. This discrepancy might be related to the inclusion criteria of each study and the type of activity evaluated. For example, carnosic acid and carnosol were used in 35 of 49 cancer studies, because they have a high antitumor activity. Rosmarinic acid extracted from *Rosmarinus officinalis* has been tested at preclinical stages to assess its anti-inflammatory and antinociceptive activity, demonstrating potential effects at these levels [1].

The plant extract was widely investigated in several clinical diseases by researchers. In some cases, specific compounds of the plant were isolated and then tested to evaluate their activity. Phytotherapy consists of plant-derived treatments where the whole plant is used to produce an extract, and its activity results from a synergic effect between the various compounds. The difference from pharmacotherapy is exactly in this synergy, because this consists of benefits from a single active substance of a drug; the same happens to an isolated compound [1].

A limitation of phytotherapeutic drugs is the natural variability of extracts. In a plant extract, the level of compounds varies, and this causes these drugs to lose biochemical consistency. Ultimately, this results in a reduction of the optimization of safety and efficacy. The natural variability can lead to inconsistent results, and this may impair the extract in being accepted as a phytotherapeutic medicine by the scientific community [1].

The reproducibility of a beneficial effect from a plant extract is greatly reduced, given the fact that multiple factors influence an extract’s activity. Some factors, such as the harvesting of the plants at different times and locations, and different extraction and quantification methods, are perhaps the reason for that limitation. The compounds isolation, purification, and structural characterization should be more profoundly developed, and for that reason, methods must be improved. Another limitation in the development of drugs from plants is that the isolation of compounds with therapeutic activity is only done in small quantities, and therefore, is not sufficient for the production of a new drug [1].

Plant extracts have been widely investigated in various clinical diseases by researchers. In some cases, specific compounds of the plant are isolated and then tested to evaluate its activity.

Not all studies reported how extracts or plant compounds were extracted because, in most of these cases, they were purchased or donated (*n* = 41). When it was possible to access this information, many different ways of extraction were mentioned, with maceration being the most prominent (*n* = 8). Afterward, the most widely used extraction methods were hydrodistillation (*n* = 7), followed by Soxhlet extraction and steam distillation (*n* = 4). Regarding maceration, the most used solvents were ethanol (*n* = 3), followed by water (*n* = 2), and both of them mixed (*n* = 2). In the case of Soxhlet extraction, the most used solvent was ethanol (*n* = 3).

In studies that used maceration as the extraction method for *Rosmarinus officinalis*, mostly dried and ground leaves were used [19,23,24,37,62]. Powder extraction was performed in a mixer with shaking [37], or slowly [62], with distilled water [19], ethanol [23,24], or both [37,62]. The extraction time varied according to the study. The temperature used was not mentioned in all studies, but the studies that noted this used room temperature [19,23,62]. The extract was filtered, and the solvent was evaporated [23,24,37,62] on a rotary evaporator [24,37].

*Rosmarinus officinalis* extracts can be obtained from several parts of the plant, such as the roots, leaves, stems, or flowers. The size of the particles influences the extraction. Thus, smaller particles are preferable, because they have more contact with the solution, which improves the extraction. Consequently, particles in the form of powder provide better extracts, because there is a much higher contact between the plant’s particles and the solvent [4]. The solvent’s temperature and pressure, as well as the extraction time, also affect the efficiency of the extraction process. In the literature, the most used extraction methods to isolate the compounds of *Rosmarinus officinalis* were maceration, hydrodistillation, distillation, and Soxhlet by supercritical fluid extraction [1]. These data are in agreement with the information contained in the studies of this review. However, contrary to the literature, in these studies, the extraction by Soxhlet was not performed using supercritical fluid extraction.

The solvent used for extraction influences which compounds are extracted, and individual extracts have a different activity depending on their compounds. The extraction method chosen will influence the final compounds in the extract, so the choice of method should take into account the properties of the plant [4].

### 2.3. Dose and Route of Administration

Given the variety of beneficial effects that *Rosmarinus officinalis* has shown, numerous in vivo animal models were conducted to test a series of doses of this plant. In the analyzed studies, several routes of administration were used, such as gavage, intraperitoneal (IP), oral (included in diet), intravenous, intrathecal, intradermal, and topical. The prevalent routes were gavage (*n* = 37), intraperitoneal (*n* = 21), and oral (*n* = 7). The effects of *Rosmarinus officinalis*, rosmarinic acid, carnosic acid, and carnosol were evaluated, and the administered doses differed according to the route. In all analyzed studies, for all doses, regardless of route and compound used, the authors had positive results.

#### 2.3.1. Gavage

Gavage (esophageal or gastric) is frequently used in research investigations to guarantee a well-defined and accurate dosing of animals, preferably combining substances with food or water. The administration of *Rosmarinus officinalis* by gavage (*n* = 18) provided an average dose of approximately 435 mg/kg, ranging between 10 and 2500 mg/kg. In one of the studies, by Faria et al. (2011), the effective dose of *Rosmarinus officinalis* was evaluated and determined to be 300 mg/kg. Furthermore, according to Takaki et al. (2008), 3000 mg/kg was determined to be the maximum dose that did not show any cases of lethality or signs of toxicity. The rosmarinic acid doses ranged between 10 and 300 mg/kg and provided an average dose of approximately 70 mg/kg, significantly higher than the other compound’s average. The carnosic acid average dose was roughly 25 mg/kg, varying within 5 and 60 mg/kg. Comparing these data, we can verify that the doses used of isolated compounds of the plant were significantly lower.

This systematic review revealed that a higher number of studies were conducted to analyze the effects of *Rosmarinus officinalis* in comparison with those to evaluate an isolated compound of the plant. Even though rosmarinic acid was more common overall, having significantly more studies than carnosic acid and carnosol. Through analyzing the data, it seems that the use of the whole plant corresponds with a necessity of higher doses, compared to studies of a concentrated substance.

Gavage presents some limitations, such as a delayed onset of the effect when compared with parenteral administration, decrease of absorption of substances, and substance degradation by digestive enzymes and acid. Furthermore, a potentially significant first-pass effect by the liver, may reduce the drug’s efficacy for the substances metabolized via this route. In this sense, the dosage through oral gavage tends to be higher [84].

#### 2.3.2. Intraperitoneal

The intraperitoneal route consists of injecting substances into the peritoneal cavity, and this is a widespread method in laboratory rodents [84]. This route was the only one used to verify the effects of the plant and all the isolated compounds mentioned earlier. The administration of rosmarinic acid (*n* = 7) and *Rosmarinus officinalis* (*n* = 6) were the most common.

Rosmarinic acid was tested in doses varying between 5 and 100 mg/kg, with an average dose of approximately 20 mg/kg. *Rosmarinus officinalis* provided an average dose of approximately 95 mg/kg, ranging between 10 and 400 mg/kg. The median lethal dose, LD50, was evaluated in one of the studies, by Faria et al. (2011), and determined to be 1000 mg/kg. Concerning carnosol (*n* = 4), its administration was mainly tested in significantly lower doses, ranging between 0.5 and 2.5 mg/kg in most studies. One particular study reported the use of 50 mg/kg. Therefore, carnosol presented an average dose of about 10 mg/kg. Last, in the case of carnosic acid (*n* = 4), the administrated doses ranged between lower values, from 5 to 60 mg/kg, giving an average dose of roughly 25 mg/kg.

In this systematic review, the number of studies regarding the evaluation of *Rosmarinus officinalis* and its main compounds through the intraperitoneal route were comparable to the gavage route. Even so, with this route, the plant and all compounds were assessed, providing a unique opportunity for comparison. Thus, comparing the doses used in each case, the use of lower doses in comparison with the gavage route was found, for both the plant and all isolated compounds. The tendency for higher doses in the usage of the whole plant seems to be transversal. It was also possible to establish that the average doses of rosmarinic acid and carnosic acid were similar, but in carnosol’s case, much lower doses were tested.

Intraperitoneal delivery is recognized as a parenteral route of administration. Parenteral administration methods usually provide the largest bioavailability. Those methods tend to evade the first-pass effect, which occurs commonly with oral administration. Therefore, in cases of intraperitoneal administration, the dosage tended to be lower in comparison with oral delivery [84].

#### 2.3.3. Oral

The administration of substances directly into the oral cavity, such as inclusion in diet (food or water), is well-established in laboratory animal experimentation. Oral administration is more economical, convenient, and moderately safe. Doses of the plant or an isolated compound given through the oral route were included in the diet, and the animals had *ad libitum* access to food. Consequently, this makes it difficult to truly evaluate the real results, because not all animals ingested the same amount of the plant or the compound under analysis.

Even so, for oral administration of *Rosmarinus officinalis* (*n* = 2), the average distributed dose was approximately 2900 mg/kg, ranging between 1250 and 5000 mg/kg. Regarding isolated compounds, both carnosic acid (*n* = 2) and rosmarinic acid (*n* = 2) were tested. Carnosic acid was distributed in doses from 5 to 20 mg/kg, with an average of nearly 12 mg/kg. In the case of rosmarinic acid, the average distributed dose was roughly 20 mg/kg, with a broader range between 15 and 200 mg/kg.

As verified for every route of administration, the use of the whole plant, as an extract, always represents a higher dosage than the administration of an isolated compound. Previous reports established the use of higher doses when studies are conducted with extracts, with lower doses of isolated constituents, when studying anti-inflammatory properties [5].

### 2.4. Frequency and Duration

There are numerous factors to take into consideration when establishing the frequency of administration in a treatment. The specific model of inflammation, route, and dose used must be considered in the decision-making process. In the analyzed studies, the doses were mainly administered daily (*n* = 40), followed by treatments as a single-dose (*n* = 31). The use of single-doses versus daily administration depends on the specific therapeutic benefits, the disease or model of inflammation, and the dose under evaluation, to fully access that treatment option.

The duration of treatment depends directly on the animal model, adapting to the conditions under analysis. In the studies in this review, the mean days that a treatment took place was 21 days, with a median of 14 days, and mode of 28 days. Therefore, from all studies reporting the duration of treatment (*n* = 42), there was a balance between treatments under (*n* = 21) and over (*n* = 21) 14 days. In the analyzed studies, treatment for 28 days (*n* = 7) was the most common.

In humans, acute inflammation is characterized by an immediate start, increasing the severity in a short time, and symptoms may persist for a few days. Subacute inflammation is the period between acute and chronic inflammation and may last 2 to 6 weeks. Therefore, a persistent inflammation that lasts for more than six weeks is considered chronic. Chronic inflammation is also associated with a slow, long-term inflammation, enduring for prolonged periods of several months to years [85]. Considering the mean life expectancy for humans versus rodents, it seems plausible to infer that in the case of rodents, chronic inflammation would develop faster, and under six weeks. Even so, the authors do not specify a timestamp in which inflammation is considered chronic for the animal. The most accurate information they give is an expectation that, with acute inflammation, it should resolve in days, or maybe a few weeks; while, chronic inflammation extends for weeks, months, or perhaps even years [86].

### 2.5. Biomarkers Evaluated

Regarding the inflammatory mediators mentioned in this review, TNF-α, IL-1β, IL-6, and myeloperoxidase (MPO), were by far the most evaluated. In all cases, an increase in these inflammatory biomarkers confirmed the onset of the inflammation. As expected, the level of those biomarkers of inflammation suffered a reduction following treatment. Interleukin 10 (IL-10), an anti-inflammatory cytokine, was also evaluated in a variety of studies, and treatment also caused the increase of this cytokine; therefore, contributing to diminishing the inflammation. These effects are evidence of the anti-inflammatory activity of *Rosmarinus officinalis,* as well as the isolated compounds analyzed. Inflammation and oxidative stress are intertwined in the numerous pathophysiological events of various diseases [5].

Regarding the oxidative stress mediators mentioned in this review, antioxidant enzymes were widely evaluated, given their connection with inflammation. In the analyzed studies, the evaluation of superoxide dismutase (SOD), catalase (CAT), reduced glutathione (GSH), and glutathione peroxidase (GPx) was taken into consideration for their possible effect on inflammatory conditions. In all the studies present in this review, a decrease of these biomarkers was verified after the induction of inflammation; consequently, manifesting as an increment in oxidative stress. As expected, the levels of antioxidant biomarkers increased following treatment, potentiating the endogenous antioxidant activity, and therefore, contributing to lessening inflammation. These effects are evidence of the antioxidant activity of *Rosmarinus officinalis,* as well as its isolated compounds.

The analyzed studies refer to several inflammatory pathways related to the anti-inflammatory activity of the plant *Rosmarinus officinalis*, as well as its main compounds. Consequently, transcription factor NF-κB was the most commonly stated (*n* = 17), followed by mentions of nitric oxide (NO) (*n* = 5) and cyclooxygenase-2 (COX-2) (*n* = 2). It has been demonstrated that rosemary extract inhibits NF-κB activation [3]. The authors report that the anti-inflammatory effect of *Rosmarinus officinalis* can be mediated by inhibition of NF-κB pathways, reducing the expression of COX-2 and inducible nitric oxide synthase (iNOS) [15,21,28,60].

Carnosol and carnosic acid, the main phenolic diterpenoid compounds of rosemary, have been noted to inhibit NO production. The inhibitory effects of carnosic acid in NO and TNF-α production are the result of the suppression of iNOS and COX-2 expression. Moreover, this inhibits the nuclear translocation of NF-κB. Carnosol attenuates the levels of iNOS and also downregulates NF-κB [87]. Moreover, this review’s findings support these data [7,63]. However, the authors state that the antioxidative and anti-inflammatory activity of either carnosic acid or carnosol alone is weaker than that of rosemary extract [87]. Other authors speculate that rosmarinic acid might exhibit an anti-inflammatory activity by inhibition of neutrophil activity, inhibition of MMP-9 activity, and modulation of the NF-κB pathway [88], accordingly to our findings [47,48,49,50,51].

Regarding inflammatory pathways, the literature reports that the transcription factors NF-κB and signal transducer and activator of transcription 3 (STAT3); inflammatory enzymes, particularly COX-2 and matrix metalloproteinase-9 (MMP-9); and last, inflammatory cytokines such as TNF-α, IL-1, IL-6, and IL-8 are the main molecular mediators of an inflammatory response. Among these mediators, transcription factor NF-κB is the principal regulator of the immune system and the inflammatory response and controls several genes encoding the cytokines, cytokine receptors, and cell adhesion molecules associated with inflammation triggering [5,89].

## 3. Materials and Methods

### 3.1. Search Strategy

Following the establishment of a review protocol based on a PRISMA methodology, the electronic databases used to search for studies were Pubmed, Scopus, and Web of Science. The keywords adapted to this study were introduced into the MeSH database to confirm if they were MeSH terms.

Depending on the database used, the terminology of the terms varied.

### 3.2. Selection of Studies

The inclusion criteria used in this systematic review included (1) inflammation, (2) presence of the plant *Rosmarinus officinalis*, and (3) studies carried out on rats or mice. The exclusion criteria used were (1) reviews, opinion articles, and clinical cases; (2) studies with exclusively in vitro models; and (3) studies with *Rosmarinus officinalis* mixed with other plants.

### 3.3. Data Extraction

The studies selected from Pubmed, Scopus, and Web of Science were analyzed by two reviewers independently. Then, the studies were compared based on various characteristics, such as animal model, plant/compound, plant/compound extraction, dose, route, frequency/duration, and biomarkers evaluated. The information in agreement was maintained, and the uneven information was reviewed by consensus. The data are presented in detail in Table 1.

## 4. Conclusions

Inflammatory diseases represent the majority of debilitating conditions. Their current therapy presents a tremendous challenge given the lack of safe, effective, and straightforward treatments. Notwithstanding that there are very effective drugs to assist in treating acute inflammation, such as steroidal and non-steroidal anti-inflammatory agents, these do not represent a viable option to treat chronic inflammation. Their regular use can be responsible for causing severe adverse reactions, including gastrointestinal, cardiovascular, and renal irregularities. Therefore, there is a great necessity to investigate further into new anti-inflammatory options with selective action and lower toxicity. Plants and their isolated compounds may represent a promising and groundbreaking source of new treatments, given their well-known anti-inflammatory and anti-oxidant activities [5].

This plant represents a potential treatment for physiological disorders, similarly or superior to the usual medications. In this review, it was possible to confirm the anti-inflammatory activity of *Rosmarinus officinalis* in several animal models, both before and after induction of treatment.

According to our review, *Rosmarinus officinalis* was mostly used in its entirety or as an extract of rosmarinic acid. *Rosmarinus officinalis* was used at a dose of 400 mg/kg via gavage and rosmarinic acid at a dose of 10 mg/kg via IP. Overall, the treatments were scheduled as daily administrations for 28 weeks. *Rosmarinus officinalis* showed anti-inflammatory activity, before and after induction treatments, with a decrease in the levels of inflammatory biomarkers and an increase of oxidative stress biomarkers.

Although the potent anti-inflammatory properties of rosemary extract have been well recognized in this review, more reliable trials are required in the future. Further evaluation of *Rosmarinus officinalis* and its main active compounds’ safety and efficacy in managing different pathological conditions is crucial.

This review may represent the first step in this revolutionary line of study and towards possible therapies.

## Figures and Tables

**Figure 1 molecules-27-00609-f001:**
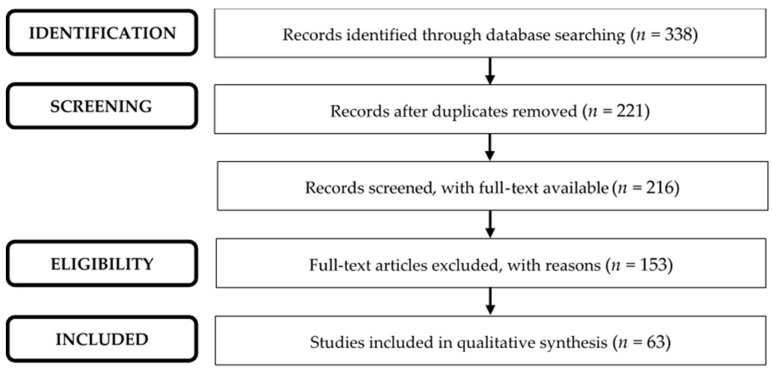
PRISMA flow diagram showing the results of the literature search.

**Table 1 molecules-27-00609-t001:** Preclinical in vivo models of inflammation using *Rosmarinus officinalis* as a therapy.

Plant/Compound	Extraction	Dose (mg/kg)	Animal Model	Route	Frequency/Duration	Biomarkers Evaluated	Reference
Carnosic acid	n.m.	10, 20	Acute liver injury	Injected	Daily, 5 d	TNF-α, IL-1β, IL-6, IL-18, IFN-γ, TGF-β, CAT, GPx, GSH, MDA, SOD	[6]
	n.a.	10–40	Acute lung injury	IP	Single-dose	TNF-α, IL-1β, IL-6	[7]
		15, 30	Non-alcoholic fatty liver disease	Oral	Daily, 8 wks	TNF-α, IL-1β, IL-2, IL-4, IL-6, IL-12, IL-18, IFN-γ	[8]
	n.m.	10, 20	Hepatocarcinoma	IP	Daily, 4 wks	TNF-α, IL-1β, IL-2, IL-6, IL-10, IL-18, IFN-γ	[9]
	n.a.	15, 30	Cirrhosis	Gavage	Daily, 8 wks	TNF-α	[10]
		5	Cardiotoxicity		Daily, 6 d	TNF-α, IL-6, COX-2, CAT, GSH, MDA, SOD	[11]
		30, 60	Arthritis	IP	Daily, 4 wks	TNF-α, IL-1β, IL-6, IL-17, IFN-γ, RANKL, MIP-1, GPx, MDA, SOD, ROS	[12]
		5	Arthritis		4 wk, 14 d	TNF-α, IL-1β, RANKL	[13]
		100, 200	Diabetes and hepatic fat accumulation	Oral	Daily, 4 wks	TNF-α, IL-6	[14]
		10, 20	Brain injury	Gavage	Daily, 9 wks	TNF-α, IL-1β, IL-6, IL-18	[15]
		15–60	Acute liver injury		Daily, 35 d	TNF-α, IL-6, GPx, GSH, MDA, SOD, NO, iNOS	[16]
		n.m.	Ear edema	Topical	Single-dose	TNF-α, IL-1β, COX-1, COX-2	[17]
*Rosmarinus* *officinalis*	Steam distillation	n.m.			Single-dose	Edema, leukocyte infiltration	
		0.6	Paw edema	Topical/Injected	Single-dose	Edema, leukocyte infiltration	
		250–750	Paw edema	Gavage	Single-dose	Edema	[18]
		125–500	Pleurisy		Single-dose	Volume of exudate, migrated cells	
	Aqueous maceration	100–400	Subcutaneous edema		Single-dose	Neutrophil infiltration, TNF-α, IL-6, PGE-2, GPx, SOD	[19]
	Aqueous maceration	150	Arthritis		Daily, 23 d	Edema, leukocyte infiltration, MPO, CAT, GPx, GSH, GSSG, GR, SOD, ROS	[20]
*Rosmarinus officinalis*	Ethanolic Soxhlet extraction	500, 1000	Acute intestinal injury	Gavage	Daily, 3 d	MPO, CAT, GSH, GSSG, MDA, SOD	[21]
		500, 1000	Gastric ulcer		Daily, 3 d	MPO, CAT, GSH/GSSG ratio, MDA, NOx, SOD	[22]
	Ethanolic maceration	50	Ear edema	ID	Single-dose	Edema, neutrophil infiltration	[23]
		100–400	Neuropathic pain	IP	Daily, 14 d	TNF-α, Iba-1, iNOS	[24]
	n.a.	50–200	Pulmonary fibrosis	Gavage	Daily, 28 d	TGF-β	[25]
		100, 300	Paw edema		Single-dose	Edema	[26]
		100	Inflammation in hippocampus		Daily, 21 d	TNF-α, IL-1β, Iba-1, NF-κB	[27]
		35, 70	Paw edema	IP	Single-dose	Edema	[28]
		0.46, 2.3	Asthma	IT	Daily	IL-5, IL-13, MIP-1	[29]
		1250–5000	Paw edema	Oral	Daily, 15 d	Edema, MPO	[30]
		1250–5000	Colitis		Daily, 18 d	IL-1β, IL-6, MPO	
		50, 100	Colitis	Gavage	Daily, 10 d	TNF-α, IL-6, MPO, NF-κB	[31]
	n.m	n.m.	Infected cutaneous wounds	Topical	Daily, 13 d	IL-3, IL-10	[32]
	Hydrodistillation	125–500	Internal spermatic fascia edema	Gavage	Single-dose	Leukocyte infiltration	[33]
		n.m.	Osteoporosis	Oral	n.m.	TNF-α, CRP, MDA	[34]
		300	Ear edema	Gavage	Single-dose	Edema	[35]
		300	Paw edema		Single-dose	Edema	
		300	Vascular permeability		Single-dose	Volume of exudate	
		300	Granulomatous		Daily, 6 d	Granulomatous tissue	
	Acetone maceration	2500	Paw edema		Single-dose	Edema	[36]
	Ethanolic maceration	2500	Paw edema		Single-dose	Edema	
*Rosmarinus officinalis*	Hydro-ethanolic maceration	10–40	Peritoneal adhesion	IP	Single-dose	TNF-α, IL-1β, IL-6, TGF-β, GSH, MDA, NO	[37]
	Methanolic Soxhlet extraction	10, 50	Paw edema	Gavage	Single-dose	Edema	[38]
Rosmarinic acid	n.a.	10–50			Single-dose	Edema	
		25	Thermal injury	IV	Single-dose	TNF-α, IL-1β, IL-6	
	Chromatography extraction	10–40	Paw edema	Gavage	Single-dose	Edema	[39]
	n.a.	100	Sepsis	IP	Single-dose	TNF-α, CAT, GPx, GSH, SOD	[40]
		5–20	Acute lung injury		Single-dose	TNF-α, IL-1β, IL-6, SOD	[41]
		5–20	Asthma	Oral	Daily, 22 d	Eosinophils/neutrophils/monocytes/lymphocytes infiltration, CAT, MDA, SOD	[42]
		50	Vascular impairment	Gavage	Daily, 10 wks	TNF-α, IL-1β, IL-6	[43]
	n.m.	5–20	Asthma	Oral	Daily, 22 d	IL-4, IFN-γ, IgE, PLA2	[44]
	n.a.	10–50	Acute liver injury	Gavage	Daily, 2 d	TNF-α, COX-2, TGF-β, SOD	[45]
	n.a.	200	Nephrotoxicity		Daily, 7 d	TNF-α, GSH, MDA	[46]
		75–300	Hepatocarcinoma		Daily, 10 d	TNF-α, IL-1β, IL-6, TGF-β	[47]
	n.m.	40, 80	Elevation of C-reactive protein		Daily, 8 wks	IL-1β, IL-18	[48]
	n.a.	75–300	Hepatocarcinoma		Daily, 10 d	IL-2, IL-6, IL-10, IFN-γ	[49]
		30, 60	Colitis		Daily, 7 d	IL-1β, IL-6, IL-22, COX-2, MPO, iNOS	[50]
	n.m.	10	Spinal cord injury	IP	Daily, 7 d	TNF-α, IL-1β, IL-6, CAT, GPx, GSH, GST, MDA, SOD, ROS	[51]
	n.a.	20	Asthma		Daily, 3 d	IL-4, IL-5, IL-13	[52]
	n.a.	5–20	Mastitis	IP	Single-dose	TNF-α, IL-1β, IL-6, MPO	[53]
		n.m.	Skin irritation	Topical	3 d	Edema	[54]
	n.m.	20	Fat graft	IP	Daily, 8 wks	TNF-α, TGF-β1, MDA	[55]
Rosmarinic acid	n.a.	10, 50	Estrogen deficiency	Gavage	Daily, 28 d	IL-18, CAT, GSH, GSSG, GSH/GSSG ratio, SOD	[56]
		10	Acute liver injury		Daily, 30 d	TNF-α, IL-6, CAT, GSH, MDA, SOD	[57]
	n.m.	50	Nephrotoxicity		Daily, 14 d	TNF-α, IL-1β, IL-6, CAT, GPx, GR, GSH, GSSH, GST, NO, SOD	[58]
	n.a.	25, 50	Neuropathic pain		Daily, 28 d	TNF-α, IL-6, MDA	[59]
	n.m.	10	Myringosclerosis		5 wk, 7 d	Edema	[60]
	n.a.	10–50	Neuropathic pain		Single-dose	IL-1β, COX2, PGE-2, NO	[61]
*Rosmarinus* *officinalis*	Ethanolic Soxhlet extraction	400		IP	Daily, 14 d	IL-1β, COX2, PGE-2, NO	
	Hydro-ethanolic maceration	10–50	Paw edema		Single-dose	COX-1, COX-2	[62]
Carnosol	n.a.	0.5, 1, 2			Single-dose	COX-1, COX-2	
		n.m.	Atopic dermatitis	n.m.	Twice	Edema, TNF-α, IL-1β, COX-2, iNOS	[63]
		50	Autoimmune encephalomyelitis	IP	Daily	IL-5, IL-10, IL-17, FN-γ	[64]
	n.m.	5	Spinal cord injury		Daily, 5 d	TNF-α, IL-1β, IL-6, CAT, GPx, GSH, GST	[65]
	n.a.	3	Acute kidney injury	IV	Single-dose	TNF-α, IL-1β, MPO	[66]
		0.0125	Atopic dermatitis	Topical	3 wk, 4 wks	Edema, TNF-α, IL-1β, COX-2, JAK, iNOS	[67]
	Chromatography extraction	2.5	Pleurisy	IP	Single-dose	Leukocyte infiltration, volume of exudate, IL-10, IL-17, MPO, NOx	[68]
*Rosmarinus* *officinalis*	Hydrodistillation	25, 50			Single-dose	Leukocyte infiltration, volume of exudate, IL-10, IL-17, MPO, NOx	
Rosmarinic acid	Chromatography extraction	5			Single-dose	Leukocyte infiltration, volume of exudate, IL-10, IL-17, MPO, NOx	

Legend: d—days; ID—Intradermal; IP—Intraperitoneal; IT—Intrathecal; IV—Intravenous; n.a.—Not applicable; n.m.—Not mentioned; wks—weeks.

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
