# Peer review of "Potential Anti-Inflammatory Effect of *Rosmarinus officinalis* in Preclinical In Vivo Models of Inflammation"

_molecules, 2022, doi:10.3390/molecules27030609_

Round 1
Reviewer 1 Report
The manuscript by Gonçalves tried to review the literature on the effect of Rosmarinus officinalis in various animal models. The structure and interpretation seemed strange and must be recognized thoroughly. The introduction is insufficient, and "results" should be deleted or combined with "discussion". Another point is the significance of this review. This reviewer could not see the direct focus of this paper, and also, no clear perspectives were presented from authors after the extensive literature reading and understanding.
Author Response
Dear Dr,
Thank you for your feedback and all the time you spent working in our manuscript.
All suggestions have been taken into account and included in the manuscript.
The English was already revised, and we already improved the manuscript, namely introduction, results/discussion and conclusions as you can verify in the manuscript.
Sincerely yours,
Prof. Vanessa Alexandra Pinho Mateus, BPharm MSc PhD
Professor of Pharmacology and Pharmacotherapy
Diretor of Pharmacy Degree Course in Lisbon School of Health Technology (Polytechnic Institute of Lisbon)
Member of Coordinating Committee of Health and Technology Research Center (H&TRC)
Reviewer 2 Report
The article with indications and the final decision it is ACCEPT AFTER MINOR REVISION, the article is an actual review about Rosmariuns officinalis and anti-inflammatory effects according the most evaluated biomarkers in vivo models of inflammation: tumor necrosis factor (TNF)-α, interleukin (IL)-1β, IL-6, 17 IL-10, myeloperoxidase (MPO), catalase (CAT), glutathione (GSH), glutathione peroxidase (GPx), 18 malondialdehyde (MDA), and superoxide dismutase (SOD).
However, I sugest few corrections and improvements, that I detailed below:
1) In table 1 reorder the columns, it is better that you include the column ANIMAL MODEL after DOSE than in the actual position. Furthermore, it will be fine if you include the concret biomarker evaluated in this column, because it is indicated as general denomination.
2) In lines 312 to 316, Rosmarinus officinalis are not in cursive letter, please correct it.
3) In conclusion section, it will be fine if a conclusion mention a clear effect detected and correlated with any of the biomarkers used.

Author Response
Dear Dr,
Thank you for your feedback and all the time you spent working in our manuscript.
The English was already revised and all suggestions have been taken into account and included in the manuscript.
Sincerely yours,
Prof. Vanessa Alexandra Pinho Mateus, BPharm MSc PhD
Professor of Pharmacology and Pharmacotherapy
Diretor of Pharmacy Degree Course in Lisbon School of Health Technology (Polytechnic Institute of Lisbon)
Member of Coordinating Committee of Health and Technology Research Center (H&TRC)
Reviewer 3 Report
The information was well organized and explained in five sections: 3.1 Animal model, 3.2 Plant/compound and extraction, 3.3 Dose and route of administration, 3.4 Frequency and duration and 3.5 Biomarkers evaluated. Each section was well structured, first by describing the statistical data of the publications related to the section and then by discussing the information of the documents. Only, section 3.2 was not finished well. It is recommended that the authors conclude the section by indicating what type of compounds are obtained in extracts or essential oils using the different extraction methods. That is, what type of compounds (examples) are best extracted by each method?
Author Response
Dear Dr,
Thank you for your feedback and all the time you spent working in our manuscript.
Suggestions have been taken into account and included in the manuscript. We already improved section Plant/compound and extraction as you can verify in the manuscript.
Sincerely yours,
Prof. Vanessa Alexandra Pinho Mateus, BPharm MSc PhD
Professor of Pharmacology and Pharmacotherapy
Diretor of Pharmacy Degree Course in Lisbon School of Health Technology (Polytechnic Institute of Lisbon)
Member of Coordinating Committee of Health and Technology Research Center (H&TRC)
Reviewer 4 Report
The authors extensively portrayed the antiflammatory activity of Rosmarinus officinalis in several animal models, with before and after induction treatments. Rosmarinus officinalis remains to be important medicinal plant in the plant kingdom with anti-inflammatory, antioxidant, antibacterial and futher beneficial effects. It is a well-known plant, used since mankind can remember.
The review contains the available amount of studies on animal models with reference to Rosmarinus officinalis.
1-The maceration and further extraction methods could have been clearer portrayed in more details.
2-These details could have included analytical results of the available components in the studies, which led to their successful use in the treatment of the animals.
3-A more definite identification of the active ingredients can be a result for different treatment methods. This is missing and makes the results/ discussion a bit vague.
4. The reference section needs to be changed in accordance to the MDPI style.
Adding more content into the results and discussion ensures the publication of this interesting manuscript.
Best of luck
Author Response
Dear Dr,
Thank you for your feedback and all the time you spent working in our manuscript.
All suggestions have been taken into account and included in the manuscript.
The English was already revised, and we already improved the manuscript, namely introduction, results/discussion, and conclusions as you can verify in the manuscript.
It was a lapse in the Reference section. Thanks for the detection.
Sincerely yours,
Prof. Vanessa Alexandra Pinho Mateus, BPharm MSc PhD
Professor of Pharmacology and Pharmacotherapy
Diretor of Pharmacy Degree Course in Lisbon School of Health Technology (Polytechnic Institute of Lisbon)
Member of Coordinating Committee of Health and Technology Research Center (H&TRC)
Round 2
Reviewer 1 Report
This reviewer has no further comments and should be acceptable in its current form.
Reviewer 4 Report
The authors answered the questions in acceptable form. The manuscript improved by adding the new parts and can be published.